# Phosphaturic Mesenchymal Tumors: Rethinking the Clinical Diagnosis and Surgical Treatment

**DOI:** 10.3390/jcm12010252

**Published:** 2022-12-29

**Authors:** Yupeng Liu, Hongbo He, Can Zhang, Hao Zeng, Xiaopeng Tong, Qing Liu

**Affiliations:** 1Department of Orthopaedics, Xiangya Hospital, Central South University, 87th Xiangya Road, Changsha 410008, China; 2National Clinical Research Center for Geriatric Disorders, Xiangya Hospital, Changsha 410008, China

**Keywords:** phosphaturic mesenchymal tumors, tumor-induced osteomalacia, hypophosphataemia, FGF-23, Tc99m-octreotide PET/CT

## Abstract

Background: The diagnosis of phosphaturic mesenchymal tumors (PMT) is easily delayed clinically, and their surgical treatment is unstandardized. This study aimed to evaluate our experience in the diagnosis and treatment of PMT and provide a research basis for the accurate and standardized treatment of PMT. Materials and Methods: Twelve patients diagnosed with PMT in our department and who underwent surgical treatment were included in this study. Preoperative demographic and clinical information were recorded. CT, MRI, and technetium-99m (Tc99m)-octreotide PET/CT imaging techniques were used to evaluate the general conditions and lesion boundaries of the tumors. Surgical treatment was performed using radical resection and microwave ablation-assisted extended curettage according to the lesion location and size. Patients were strictly followed up with and evaluated for oncological prognosis, radiological results, bone healing, serum ion levels, limb function, and pain level; the occurrence of complications was also recorded. Results: Three patients underwent radical resection, and nine underwent microwave ablation-assisted extended curettage. The average duration of symptoms in this group was 1.5 years (9–35 months) before diagnosis. Serum phosphate and AKP levels returned to normal one and two weeks postoperatively, respectively. There was no apparent specificity in the pathological findings; however, the immunohistochemistry of FGF-23 was positive, and the original fracture sites were effectively healed during the follow-up. The limb function and pain scores were significantly improved. The MSTS score increased from 15.3 to 29.0, and the VAS score decreased from 5.3 to 0.4. All patients recovered, and 90% resumed their original jobs. Conclusions: Accurate diagnosis and standardized surgical treatment are crucial to achieving a clinical cure for PMT. Combining clinical manifestations, biochemical examinations, imaging characteristics, and pathological findings is an effective way to diagnose PMT accurately. Radical resection and microwave ablation-assisted extended curettage are reliable surgical treatment methods for PMT.

## 1. Introduction

Tumor-induced osteomalacia (TIO), also known as tumor-induced osteomalacia, is an extremely rare paraneoplastic syndrome characterized by hyperphosphaturia, hypophosphatemia, and elevated alkaline phosphatase levels [1,2,3,4]. Phosphaturic mesenchymal tumors (PMT), the primary tumor type causing TIO, are rare tumors originating from mesenchymal tissues, such as bone and soft tissues [5,6,7,8,9]. The clinical manifestations of PMT are usually generalized bone pain, osteoporosis or osteomalacia, and in severe cases, multiple fractures and movement disorders [5,9,10].

The pathogenesis of PMTs is still unclear and is primarily related to the abnormal tumor secretion of fibroblast growth factor 23 (FGF-23) [1]. FGF-23 affects phosphate reabsorption and vitamin D metabolism by regulating the proximal renal tubular sodium phosphate cotransporters II and affecting the 25-hydroxyvitamin D3 1-alpha-hydroxylase function, resulting in decreased blood phosphate, increased urine phosphate, and impaired bone mineralization [1,11,12]. Therefore, in the pathological diagnosis of PMT, we often need to distinguish it from a non-ossifying fibroma, fibrosarcoma, osteosarcoma, osteoblastoma, sclerosing hemangioma, angiofibroma, or angiolipoma [13,14,15]. Studies have shown that FGF-23 and somatostatin receptor 2A (SSTR2A) are highly sensitive but nonspecific for a PMT diagnosis [13,16,17]. Therefore, negative staining of these two markers can be used to exclude PMT. The latest research results suggest that FGFR1 immunohistochemistry and FN1-FGFR1 fusion gene detection can further diagnose PMT [18].

Clinically, patients with PMT often present with progressive and worsening systemic bone pain, osteoporosis, or osteomalacia, which are usually the result of chronic hypophosphatemia rather than directly relating to the tumor itself [16,17]. The clinical presentation can be dramatic, with previously healthy people complaining of progressive weakness, activity limitations, skeletal deformities, fractures, and false fractures that eventually affect mobility and sometimes confine patients to wheelchairs. Due to the low incidence of PMT, its clinical manifestations are incredibly confusing and difficult to diagnose, leading to a high misdiagnosis rate and making it easily ignored. Previous studies, including sporadic case reports and a small number of case series, focused more on its pathological characteristics [15,19,20,21]. Similarly, there are few extensive data reports on its clinical diagnosis and surgical treatment [6,22], leading to the lack of a reliable surgical method for its clinical treatment.

The present study aimed to expand the understanding of this rare condition and share relevant experience from our bone tumor center concerning its preoperative diagnosis and surgical treatment to provide a research basis for its standardized treatment, achieve a rapid and accurate diagnosis, and optimize its treatment strategy. 

## 2. Materials and Methods

### 2.1. Patients

Patients with PMT treated at the bone tumor center of Xiangya Hospital between June 2015 and June 2020 were included. The study was conducted according to the Declaration of Helsinki and approved by the Research Ethics Committee of Xiangya Hospital. Written informed consent was obtained from patients or their legal guardians. The inclusion criteria were as follows: PMT diagnosis, patients who underwent surgical treatment, and continuous follow-up of >24 months. The exclusion criteria were as follows: PMT not confirmed on pathological examination, nonsurgical treatment, and <12 months of follow-up or incomplete follow-up data.

The cohort consisted of 12 patients who met the inclusion criteria, including five males and seven females, with an average age of 41.7 ± 14.9 years. The patient’s age, gender, tumor location, tumor size, disease duration, presence or absence of pathological fractures, preoperative limb function score, preoperative pain score, bone metabolism-related hormone levels, and preoperative serum calcium and phosphate concentrations were recorded (Table 1). Limb function evaluations were performed according to the International Society of Limb Salvage and the Musculoskeletal Tumor Society (MSTS) scoring system [23]; the pain was graded using the visual analogue scale (VAS) [24].

### 2.2. Diagnosis

All patients were preoperatively examined using radiography, computed tomography (CT), magnetic resonance imaging (MRI), radionuclide bone imaging, and technetium-99m (Tc99m)-octreotide positron emission tomography (PET)/CT scans to evaluate the general condition and lesion boundaries. We obtained preoperative clinical and biochemical characteristics, including serum ion concentrations and hormone levels related to bone metabolism, particularly the serum phosphate ion concentration, alkaline phosphatase (AKP) level, and parathyroid hormone (PTH) level (Appendix A). Ten patients underwent preoperative puncture biopsies; histological examinations were performed independently by two experienced pathologists. Moreover, our bone tumor center’s multi-disciplinary team (MDT) consulted on all cases in this study. The final clinical diagnosis was obtained by combining the patients’ clinical manifestations and imaging and pathological features. 

### 2.3. Pre-Operative Evaluation

We used radical resection and microwave ablation-assisted extended curettage for treatment according to the lesion location and range of involvement. The authentic boundary of the lesion was determined according to the T1WI-enhanced MRI images. For patients with bone lesions who underwent radical resection, tumor-type artificial joints (WEGO, Shandong, China) were used for the repair and reconstruction (Figure 1). For patients undergoing microwave ablation-assisted extended curettage, we first used microwave ablation to inactivate the lesions (50 W, 2 min) (Figure 2); afterward, we used different types of curettes to expand the lesions’ curettage. A high-speed grinding drill was used to expand the boundary of the tumor cavity. Next, bone cement or allogeneic bone was used to repair the bone defect and assist the internal fixation devices (Stryker, Michigan, USA). Lesions located in the soft tissue were treated using radical resection. Finally, the incision was sutured successively, and a drainage tube was placed. We intravenously administered antibiotics until the drainage tube was removed to encourage early postoperative activity.

### 2.4. Postoperative Management

Physiotherapists instructed patients to use braces and walk with crutches to contract the affected muscles. Isometric contraction exercise was initiated one week postoperatively, and passive functional exercise was initiated two weeks postoperatively, under the guidance of physiotherapists. Patients gradually transitioned from no-weight to half-weight to full-weight bearing. The patients’ serum ion levels were reviewed every three days postoperatively for two weeks; the bone metabolism-related hormone levels were reviewed simultaneously. Patients were followed up with at 6 and 12 weeks postoperatively and then at three-month intervals for the first two years, every six months for the next three years, and annually thereafter. The follow-up primarily evaluated the patients’ oncological prognosis, radiological results, bone healing, serum ion levels, limb function, and pain levels and recorded the occurrence of complications.

### 2.5. Statistical Analysis

SPSS v. 26.0 (SPSS Inc., Chicago, IL, USA) was used for statistical analysis. The measurement data are expressed as mean ± standard deviation. Descriptive statistical analysis was performed for the entire study group. Paired *t*-tests were used to evaluate the preoperative and postoperative follow-up results, and *p*-values < 0.05 indicated statistical significance. 

## 3. Results

### 3.1. Clinical Information

All 12 patients were followed up with for an average of 46.2 months. Of these, 10 patients had lesions in the bone, and 2 had lesions in the soft tissues. All patients complained of a long-term history of osteoporosis, generalized fatigue, pain, fractures or false fractures, and weakness. Four patients localized the pain to the tumor site, eight showed TIO-related pain outside the primary focus, and nine had pathological fractures. Seven patients received medical treatment, such as anti-osteoporosis and phosphate supplementation, before being diagnosed with PMT by our center’s MDT. Five patients underwent surgical treatment, including one who remained undiagnosed with PMT after multiple surgical treatments. The average duration of symptoms in this group was 1.5 years (9–35 months) before diagnosis (Table 1). 

### 3.2. Imaging, Biochemical, and Pathological Characteristics

In this cohort, preoperative Tc99m-octreotide PET/CT scans showed apparent hypermetabolic concentrations at the lesion sites, indicating that somatostatin receptor expression was abnormally high. All patients had significant hypophosphatemia and elevated AKP levels preoperatively. Some patients also had elevated PTH and total type I collagen N-terminal lengthening peptide (TPINP) levels. However, these patients’ serum calcium levels were within the normal range. The biochemical tests showed that the serum phosphate concentration increased gradually after the operation and returned to normal approximately one week postoperatively, with significantly different levels from preoperative levels. In addition, AKP and PTH levels decreased significantly postoperatively. Most patients recovered to normal or slightly higher levels at approximately two weeks postoperatively, showing significant differences from preoperative levels. The pathological characteristics of this study cohort were mainly that PMT was composed of the hypocellular proliferation of bland neoplastic spindled cells growing in a highly vascular, hyalinized, partially calcified, basophilic matrix. However, individual PMTs had significant differences in cellular, vascular, and matrix composition. Therefore, the morphological spectrum of PMT was quite extensive. Meanwhile, the immunohistochemical results for FGF23 and SSTR2A in this cohort were positive, while the results for SATB2, ERG, CD56, and S-100 were inconsistent (Figure 3).

### 3.3. Oncology Prognosis

This cohort was treated with radical resection or microwave ablation-assisted extended curettage; no evidence of recurrence or distant metastasis was found during postoperative follow-up. Two patients with lesions located in soft tissues, one in the femur and one in the scapula, underwent radical resection. Patients with lesions located in the femur (n = 4), tibia (n = 1; Figure 4), humerus (n = 1), and ilium (n = 2; Figure 5) were treated with microwave ablation-assisted expanded curettage and bone cement filling with an internal fixation device. One case was treated with tumor segment resection and tumor-type artificial joint replacement in the proximal femur. Pathological fractures occurred in nine patients preoperatively, including three multiple fractures, four fractures located in the lesion, and two limb fractures outside the lesion. Before the primary lesion was treated, there was no sign of healing at the fracture site. However, after treating the primary tumor site, the follow-up showed that the original fracture site healed effectively. All patients in this cohort had significant pain and limb activity limitations preoperatively, with a mean preoperative MSTS score of 15.3 and a mean preoperative VAS score of 5.3. Among them, three patients used a wheelchair, three were admitted with crutches, and one was confined to a bed for two years. In the last postoperative follow-up, patients reported that the pain had disappeared, the limb weakness and fatigue were significantly improved compared with preoperative levels, the original fracture site had healed, and limb function had recovered. The average postoperative MSTS score was 29.0, which was significantly different from the preoperative score (Table 2). All patients returned to their normal lives, and 90% resumed their original jobs.

## 4. Discussion

Hypophosphorous osteomalacia (HPO) is a metabolic bone disease in which the newly formed bone matrix (osteoid or osteoid tissue) cannot complete mineralization normally [25,26,27]. There are two types of HPO: inherited and acquired. In inherited diseases, autosomal recessive, X-linked, and autosomal dominant hypophosphatemia are caused by a gene mutation that leads to the abnormal production of FGF-23 [26]. HPO diagnosed in adults is mostly acquired. These diseases include vitamin D deficiency, renal failure, drug-induced osteomalacia, and TIO. As the primary TIO type, PMTs are rare tumors that ectopically secrete FGF-23 [1,11]. FGF-23 is an osteocyte-derived protein that regulates phosphate homeostasis. Overproduction of FGF-23 leads to a paraneoplastic syndrome where FGF23 inhibits 1,25 hydroxylase (CYP27b1) function and stimulates CYP24, therefore decreasing synthesis and increasing the catabolism of 1,25-dihydroxyvitamin D [11,12]. 

PMT primarily occurs in adults, without noticeable sex differences, and is primarily found in bone, soft tissue, and skin, among which the bone ends of the limbs are the most common locations. Because the tumors are often small and hidden, the time from symptom onset to tumor resection often ranges from months to years. Typical clinical symptoms are bone pain, deformity, limited activity, and muscle weakness that gradually worsen with the disease progression [5,9,14,28]. The pathology of PMT lacks specificity. It primarily comprises obese spindle cells, osteoclast-like multinucleated giant cells, abundant blood vessels, mature adipose tissue, osteoid tissue, floccular calcified chondroid matrix, and bleeding [13,18]. Among them, spindle cells express FGF-23 and may be considered tumor parenchymal cells. Although most PMTs appear benign, tumors may occasionally exhibit malignant features and aggressive clinical behavior [20,29]. Due to the rarity of PMT, most existing studies are case reports and pathological feature series studies [13,15,30]. There is a lack of large-scale clinical treatment research reports on PMT. Based on clinical practice, this study shares our experience concerning PMT diagnosis and surgical treatment to provide data support and a research basis for the accurate diagnosis and standardized treatment of PMT.

The correct diagnosis of PMT is the basis for its correct treatment. In this cohort, the average disease duration was 25 months, which was not unrelated to early misdiagnosis or missed diagnosis. All the patients received medical or surgical treatment in other hospitals, which was closely related to the clinicians’ lack of understanding of PMT. According to our experience, when patients present with refractory hypophosphatemia with clinical manifestations of osteomalacia, HPO should be considered first. HPO is mainly caused by endocrine or genetic factors. Considering that the patient has acquired onset and progressive aggravation, it is more likely to be caused by endocrine abnormalities. Therefore, we should consider endocrine tumor factors. When the patient’s imaging examination has no specific performance, we can consider using the diagnostic technology used for neuroendocrine tumors, Tc99m-octreotide PET/CT scans [2,6,31,32], to screen for the responsible tumors.

Similarly, the pathological examination of puncture biopsies facilitates diagnosis. Pathological manifestations of nonspecific tumor-like changes, accompanied by the positive expression of FGF-23 [10,13,17], increase the odds of diagnosing PMT. Therefore, we made a comprehensive diagnosis of PMT based on the biochemical characteristics of refractory hypophosphatemia, the imaging results of Tc99m-octreotide PET/CT scans, and FGF-23-positive expression on histopathology, after consulting with the MDT team (Figure 6).

PMT is a paraneoplastic syndrome, a systemic symptom caused by local neoplastic lesions [33,34]. Therefore, the localization and qualitative diagnosis of the responsible tumors are crucial. A qualitative diagnosis primarily depends on the characteristic clinical manifestations, including chronic progressive systemic bone pain and limb weakness. Biochemical examination showed low serum phosphate, elevated AKP, and normal serum calcium and PTH levels. X-ray examinations and bone mineral density measurements indicated osteomalacia. Osteomalacia caused by other factors, such as malnutrition, liver and kidney diseases, and family-inherited diseases, should also be excluded.

Regarding a localization diagnosis, the Tc99m-octreotide PET/CT scan is the preferred imaging examination; however, there are also false-negative and false-positive results. For patients with difficult localizations and high suspicion of PMT, 68Ga-DOTA-TATE-PET/CT can be performed, which can be localized and qualitative [31,35,36]; however, it is expensive, and few hospitals can conduct this examination. CTs and MRIs can detect and visualize lesions more clearly and help define the boundaries of surgical resections; however, they are unsuitable for finding the responsible tumors. Since PMTs are most frequently found in bone tissues, it is often necessary to differentiate them from brown tumors, giant cell tumors of bone (GCTB), osteosarcoma (OS), and chondroblastoma in clinical practice [5,8]. Brown tumors occur in females and are often caused by parathyroid adenomas. They are characterized by multiple localized cystoid bone destruction, boundary clearing, elevated serum calcium levels, and decreased serum phosphate levels. The tumors can self-heal after adenoma resection. GCTBs occur at the bone ends, presenting with expansive multilocular eccentric bone destruction without internal calcification or ossification shadows. OSs are most common in adolescents and occur in the metaphysis of long bones, with characteristic tumor osteogenesis, Codman’s triangle, and soft tissue masses. Chondroblastomas occur at secondary ossification centers, presenting as eccentric cystic expansions of osteolytic destruction with scattered calcification shadows within the lesions. In addition, except for brown tumors, the other three tumor types generally do not have systemic osteomalacia symptoms and histories of hypophosphatemia.

There have been no clear guidelines or reference documents for treating PMT. Previous studies on the surgical treatment of PMT are relatively rare [6,22,37]. A few studies have shown a high recurrence rate after curettage treatment [36], while others reported distant metastasis [10,29]. Currently, the core of clinical treatment for patients with a definite diagnosis of PMT is the complete resection of the responsible tumor. Once the responsible tumor is removed, the patient’s systemic symptoms and blood biochemical indexes improve and reach the normal range. However, there are also sporadic case studies on treating PMT with vitamin D, calcium agents, and FGF-23-targeted drugs. 

In this study, patients with PMT were treated by radical resection and microwave ablation-assisted extended curettage. Fortunately, no signs of tumor recurrence were found during follow-up, and all patients were cured. Patients’ serum phosphate ion and AKP levels returned to normal; x-rays showed that patients’ bone mineral density was significantly increased, the original fracture site was healed, and limb function was restored to a normal state. Radical resection is a very effective surgical method that can completely remove the lesion from the normal tissue with very low postoperative recurrence rates. This procedure is primarily applied to patients with lesions located in soft tissues or within whom a large extent of the lesions involves bones that are difficult to fix with internal fixation devices. Microwave ablation-assisted extended curettage involves using microwave ablation to inactivate the lesion in a large range, extending the curettage treatment, and finally reconstructing the bone defect. This procedure is suitable for patients with small bone lesions that can be effectively fixed. Considering most lesions are located in the bone and the scope is relatively limited, radical resection is bound to cause significant bone defects and is not the first choice for a benign tumor. Therefore, using microwave ablation-assisted extended curettage is more reasonable. Due to the gross intraoperative complexity of PMT lesions, which comprises bony or fibrous tissue, it is difficult to distinguish the real boundary of the tumor; therefore, we first inactivated the lesions and surrounding bone tissue using microwave ablation, thus providing a safe curettage boundary. Afterward, conventional expanded curettage was used to treat the tumor cavity, ensuring intraoperative tumor removal. Finally, the bone defect was repaired by bone grafting or bone cement filling, assisted by an internal fixation device, to maximize the preservation of limb function and achieve precise treatment of PMT.

Previous studies reported many cases of delayed PMT diagnosis and relapse after treatment; some patients even had distant metastases [10,20,29,36]. This is closely related to the different clinical manifestations and typical occult symptoms of PMT. Similarly, the inexperience of clinicians and nonstandard treatment also bear unshirkable responsibilities. Based on previous studies and the results of this study, accurate diagnosis and standardized treatment of PMT are crucial to achieving its clinical cure. Through this study, we identified the following core elements. First, the abnormality of clinical symptoms and biochemical indicators is related to tumor-related diseases, and simple symptomatic treatment is often futile. Second, the reasonable selection of imaging technology facilitates the qualitative and localization diagnosis of lesions; Tc99m-octreotide PET/CT is the ideal imaging technology. Third, histological examination and staining for specific markers can provide the most powerful evidence for diagnosis. In addition, FGF-23 is the most sensitive screening marker. Fourth, MDT consultation is the most authoritative and standardized PMT diagnosis method. Finally, standardized surgery fundamentally guarantees the achievement of a clinical cure and reduces postoperative recurrence rates.

This study reported on a clinical research cohort for PMT in our center and shared our relevant experience in the clinical treatment of PMT with respect to preoperative diagnosis and surgical treatment. However, this study was limited by the small sample size and limited follow-up time. A long-term, multicenter follow-up study with a large sample size should be conducted.

In conclusion, accurate diagnosis and standardized surgical treatment are crucial to clinically curing PMT. The combination of clinical manifestations, biochemical examinations, imaging characteristics, and pathological findings is effective for accurately diagnosing PMT. Radical resection and microwave ablation-assisted extended curettage are reliable surgical treatments for PMT.

## Figures and Tables

**Figure 1 jcm-12-00252-f001:**
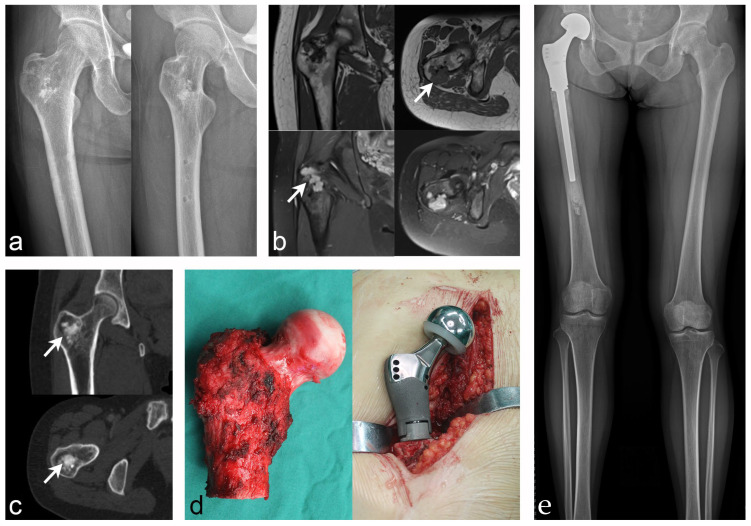
Case 4: PMT of the proximal femur that underwent radical resection and tu-mor-type artificial joint replacement. (**a**) Anteroposterior and lateral X-rays of the proximal femur revealed intertrochanteric bone abnormalities. (**b**) MRI showed low intensity on T1WI and high intensity on T2WI (white arrow indicates the lesion). (**c**) CT showed heterogeneous lesion density with osteogenic changes (white arrow indicates the lesion). (**d**) Tumor segment resection and tumor-type artificial hip replacement were performed. (**e**) Postoperative X-ray showed that the prosthesis was stable and the lower limbs were basically equal in length.

**Figure 2 jcm-12-00252-f002:**
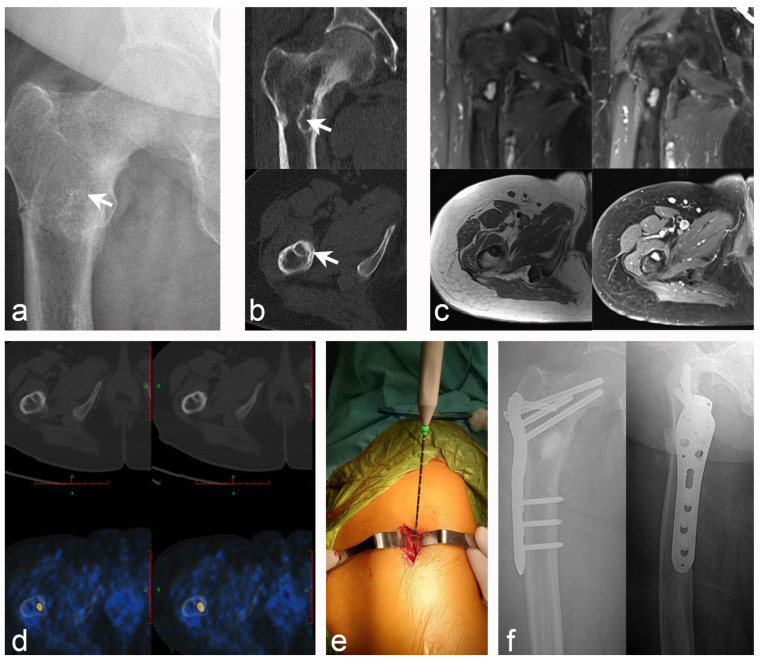
Case 10: PMT of the right proximal femur with pathological fracture; microwave ablation-assisted extended curettage was performed, and bone graft filling was assisted by an in-ternal fixation device. (**a**) Anteroposterior X-ray of the proximal right femur (white arrow indicates the lesion). (**b**) CT scan revealed an osteolytic lesion with a marked sclerosing margin below the lesser trochanter of the right femur (white arrow indicates the lesion). (**c**) MRI showed low intensity on T1WI and high intensity on T2WI, with obvious en-hancement after lipid suppression enhancement. (**d**) Tc99m-octreotide PET/CT scan accurately located the lesions and showed abnormal concentration. (**e**) Intraoperative details: microwave ablation was used to inactivate the lesions at the beginning. (**f**) Postoperative X-rays of the an-teroposterior and lateral after extensive curettage of the lesion with bone graft filling and proximal femoral anatomical plate fixation.

**Figure 3 jcm-12-00252-f003:**
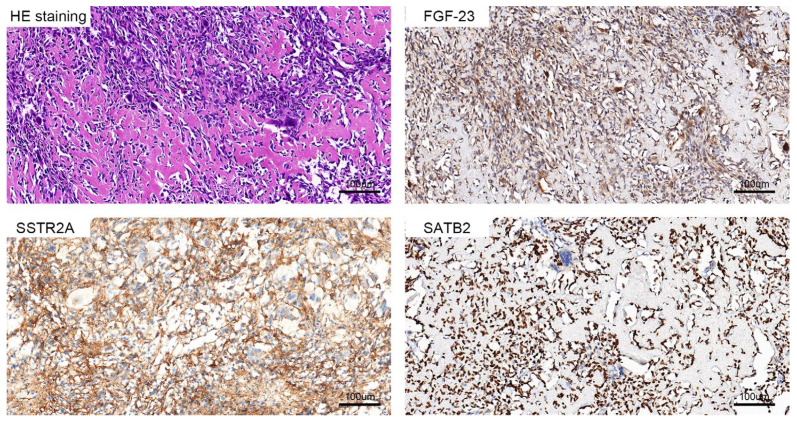
The results of the pathological examination of PMT showed that the staining was mostly spindle cells and bone-like matrix components, and the immunohistochemical staining of FGF23, SSTR2A, and SATB2 was strongly positive.

**Figure 4 jcm-12-00252-f004:**
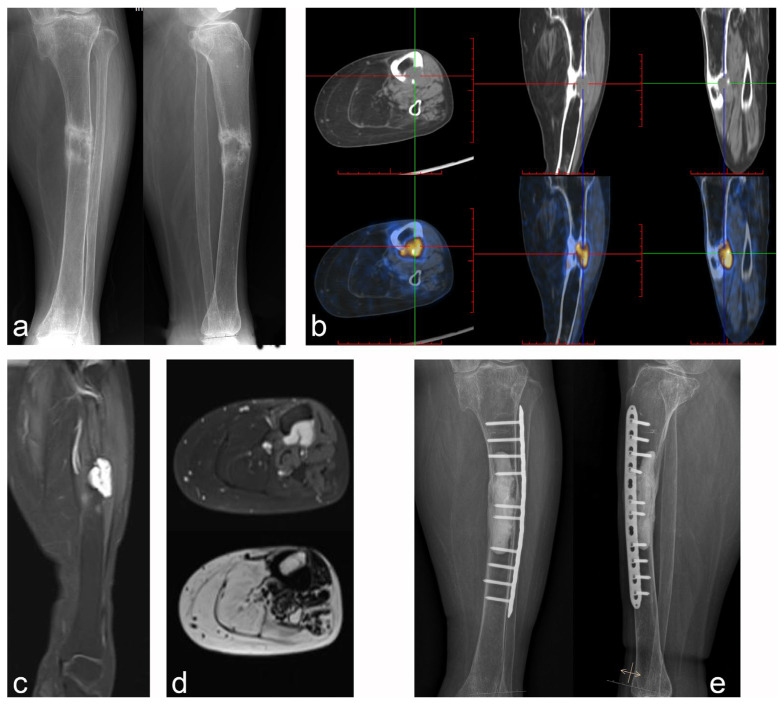
Case 5: PMT of the middle tibia with pathologic fracture underwent microwave ablation-assisted extended curettage. (**a**) Anteroposterior and lateral X-rays showed osteolytic lesions in the middle tibia with pathological fractures and obvious hardening of the fracture ends. (**b**) Tc99m-octreotide PET/CT scan showed an abnormal concentration of the lesion. (**c**,**d**) MRI showed low intensity on T1WI and high intensity on T2WI, with obvious enhancement after lipid suppression enhancement. (**e**) Microwave ablation-assisted extended curettage was performed, followed by cement filling and assisted by an internal fixation device.

**Figure 5 jcm-12-00252-f005:**
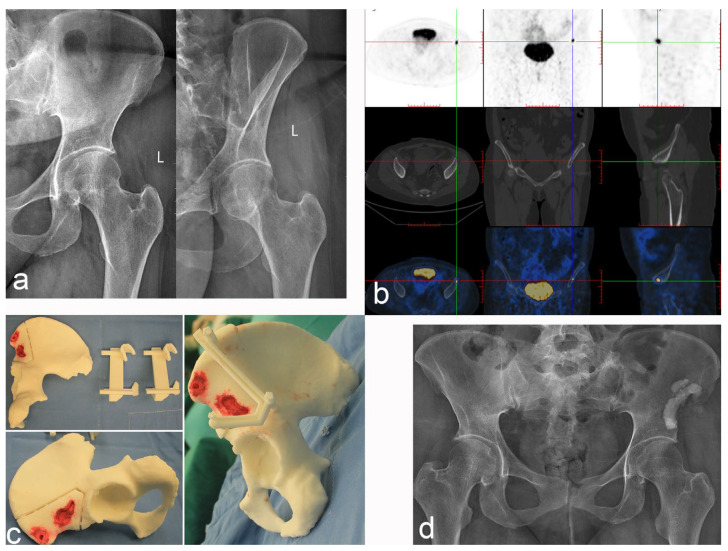
Case 2: PMT of the left iliac crest was treated with radical resection and cement filling. (**a**) Anteroposterior and oblique X-rays of the pelvis. (**b**) Tc99m-octreotide PET/CT scan accurately located the lesions and showed abnormal concentration. (**c**) The 3D printing technology reproduces the lesion and designs the osteotomy guide plate. (**d**) Postoperative anteroposterior X-ray of the pelvis.

**Figure 6 jcm-12-00252-f006:**
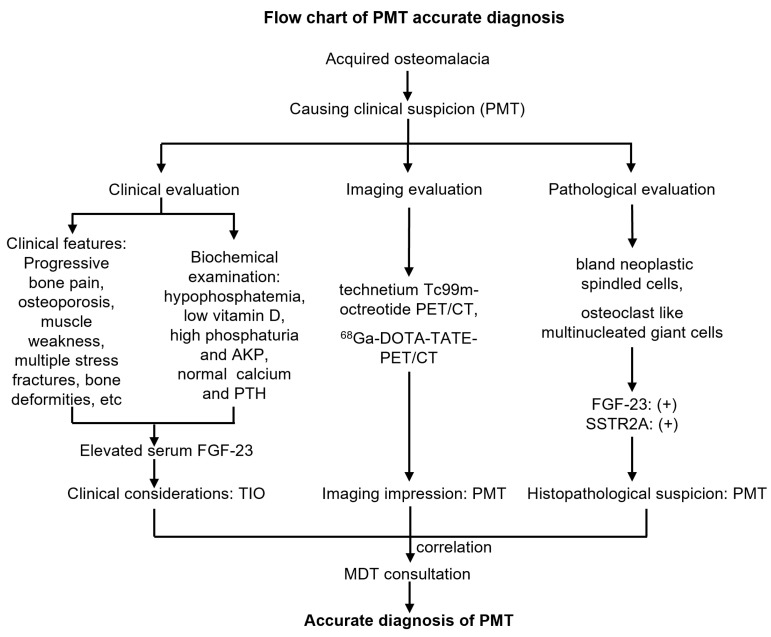
Flow chart of accurate diagnosis of phosphaturic mesenchymal tumors.

**Table 1 jcm-12-00252-t001:** Demographic and clinical information of patients.

General Information		Mean	SD
Age		41.7	14.9
Duration of disease (month)		21.2	8.8
Tumor size (cm)		3.8	1.5
Duration of Follow-up (month)		46.2	14.1
**General Information**		**Number**	**Percentage**
Gender	M	5	41.7%
	F	7	58.3%
Location	Femur	5	41.7%
	Tibia	1	8.3%
	Humerus	1	8.3%
	Scapula	1	8.3%
	Ilium	2	16.6%
	Soft tissue	2	16.6%
Treatment before diagnosis	Surgery	5	41.7%
	Medication	7	58.3%
Pathological fracture	Yes	9	75%
	No	3	25%
Therapeutic method	MAAEC	8	66.7%
	RR	4	33.3%

MAAEC, microwave ablation-assisted extended curettage; RR, radical resection.

**Table 2 jcm-12-00252-t002:** Comparative statistical analysis of various factors.

Comparative Analysis	Stage	Mean ± SD	*p* Value
MSTS score	Preoperative	15.3 ± 2.8	<0.001
	Postoperative	29.0 ± 0.9	
VAS score	Preoperative	5.3 ± 1.0	<0.001
	Postoperative	0.4 ± 0.5	
Phosphate	Preoperative	0.4 ± 0.1	<0.001
	14 days of postop	1.3 ± 0.1	
AKP	Preoperative	284.7 ± 124.6	<0.001
	14 days of postop	93.8 ± 21.9	
PTH	Preoperative	103.0 ± 129.0	0.25
	Postoperative	59.3 ± 8.9	
Calcium	Preoperative	2.2 ± 0.1	0.32
	Postoperative	2.2 ± 0.1	
TPINP	Preoperative	101.4 ± 52.0	<0.05
	Postoperative	66.2 ± 18.8	

MSTS, Musculoskeletal Tumor Society; VAS, visual analogue scale; AKP, alkaline phosphatase; PTH, parathyroid hormone; TPINP, type I collagen N-terminal lengthening peptide.

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
