# Peer review of "Phosphaturic Mesenchymal Tumors: Rethinking the Clinical Diagnosis and Surgical Treatment"

_jcm, 2022, doi:10.3390/jcm12010252_

Round 1

Reviewer 1 Report

This is a retrospective unicentric case series of phosphaturic mesenchymal tumors, including their surgical management and follow-up.

Taking into consideration the rarity of the tumors, this is an informative and interesting article which adds to current knowledge about PMTs. Also, the paper is well-written and the figures very clear and usefule. My recommendations in order to improve the manuscript:

1. Introduction: The terminology reagrding PMTs is not presented in a clear way. Which is the tumor and which is the paraneoplastic syndrome? For example, you write: 

The diagnosed tumor types include giant bone cell tumor, non-ossifying fibroma, fibrosarcoma, osteosarcoma, osteoblastoma, sclerosing hemangioma, angiofibroma, angiolipoma, or other mesenchymal tumors

Are all these tumor types PMTs? In my opinion, you should present the WHO classification about PMTs in order to be precise and clear.

2. The title of 2.2 Diagnosis could be changed to Pre-operative evaluation

3. The title of Table 2 should be Comparative statistical analysis between pre-operative and post-operative variables. You should also analyze briefly in the text the results of these correlations. Which results were anticipated and which not? How do you explain the results?

Minor: Minor grammar changes should be made.

Author Response

Dear Editors and Reviewers,

On behalf of my co-authors, we thank you very much for giving us an opportunity to revise our manuscript, we appreciate editors and reviewers very much for their positive and constructive comments and suggestions on our manuscript entitled “Phosphaturic mesenchymal tumors: rethinking from the perspective of clinical diagnosis and surgical treatment”. We have studied reviewers’ comments carefully. Those comments are all valuable and very helpful for revising and improving our paper, as well as the important guiding significance to our study. We have tried our best to revise our manuscript according to the comments, which we hope meet with approval. Revised portion are marked in red in the paper. The main corrections in the paper and the responds to the reviewers’ comments are as follows.

Comments to the Author

Reviewer: 1

  1. Introduction: The terminology reagrding PMTs is not presented in a clear way. Which is the tumor and which is the paraneoplastic syndrome? For example, you write: The diagnosed tumor types include giant bone cell tumor, non-ossifying fibroma, fibrosarcoma, osteosarcoma, osteoblastoma, sclerosing hemangioma, angiofibroma, angiolipoma, or other mesenchymal tumors. Are all these tumor types PMTs? In my opinion, you should present the WHO classification about PMTs in order to be precise and clear.

Answer: Thank you very much for taking time out of your busy schedule to review this manuscript. Tumor-induced osteomalacia (TIO), also known as oncogenic osteomalacia, is a rare paraneoplastic syndrome characterized by abnormal phosphate and vitamin D metabolism (abnormal phosphate homeostasis) due to renal phosphate wasting. Phosphaturic mesenchymal tumors (PMT) is a rare tumor originating from mesenchymal tissue (such as bone and soft tissue). Its histological morphology is diverse, and its main histological type is mixed connective tissue cell type. From histology alone, PMT is easily misdiagnosed as giant bone cell tumor, non-ossifying fibroma, fibrosarcoma, osteosarcoma, osteoblastoma, sclerosing hemangioma, angiofibroma, angiolipoma, or other mesenchymal tumors. PMT, although challenging to diagnose, is a well-defined pathologic entity, driven by gene fusions involving FGF1 or FGFR1 in up to 50% of cases. It is typically composed of cytologically bland fibroblasts with a prominent vascular network, and amorphous, "grungy" calcifications. The latest histological classification does not make any classification of PMT subtypes, so we usually make an accurate diagnosis of patients according to their clinical manifestations, biochemical tests and pathology. We have revised the content of the manuscript according to your professional suggestions. Thank you again for your valuable suggestions.

  1. The title of 2.2 Diagnosis could be changed to Pre-operative evaluation

Answer: Thank you for your professional suggestion. We have accepted your professional suggestion and changed the title of 2.2 Diagnosis to Pre-operative evaluation.

  1. The title of Table 2 should be Comparative statistical analysis between pre-operative and post-operative variables. You should also analyze briefly in the text the results of these correlations. Which results were anticipated and which not? How do you explain the results?

Answer: Thank you for your professional suggestion. We have accepted your professional suggestion and changed the title of Table 2 to Comparative statistical analysis between pre-operative and post-operative variables. At the same time, we briefly analyzed the differences of biochemical indexes before and after operation in the manuscript. According to our treatment experience, the most accurate manifestation of effective surgical treatment for PMT patients is that the level of calcium and phosphorus can be restored to the normal level within one week after surgery. Meanwhile, markers related to bone metabolism, such as AKP, PTH and TPINP, will also change significantly, which is in line with the basic criteria of human bone metabolism and the balance of calcium and phosphorus. When we realize that PMT is a kind of endocrine tumor, we can explain the abnormal biochemical indexes caused by it. We have revised the content of the manuscript according to your professional suggestions. Thank you again for your valuable suggestions.

Once again, thank you very much for editor’s and reviewers’ positive and constructive comments. Your suggestions make us progress continuously. We hope that our studies will provide further experience and evidence for clinical diagnosis and surgical treatment of PMT.

Reviewer 2 Report

This is an interesting case series describing the outcomes of surgical excision and ablation of phosphaturic mesenchymal tumor. Although is it well known that complete removal of the tumor typically leads to resolution of osteomalacia, the rarity of this disease makes the study worthwhile. However, there are multiple issues with this manuscript:

1. I have major issues with the histologic description of these cases. It appears that a Pathologist with experience in bone and soft tissue tumors was not involved in this study (and if a Pathologist made these diagnoses, which they did, they should be included in the study).
PMT, although challenging to diagnose, is a well-defined pathologic entity, driven by gene fusions involving FGF1 or FGFR1 in up to 50% of cases. It is typically composed of cytologically bland fibroblasts with a prominent vascular network, and amorphous, "grungy" calcifications. The authors' statement saying "This cohort's pathological findings lacked specificity, showing different histocytological morphology, such as giant cell-change-rich fibrous tissue, cell-rich angiofibroma, non-ossifying fibroma, and osteoblastoma" is problematic. It is true that PMT may have some histologic overlap with these entities, but that does not make its histologic features non-specific. Similarly, the statement "The diagnosed tumor types include giant bone cell tumor, non-ossifying fibroma, fibrosarcoma, osteosarcoma, osteoblastoma, sclerosing hemangioma, angiofibroma, angiolipoma, or other mesenchymal tumors" is simply not true. The authors need to more carefully read references 13-15. 

2. Besides immunohistochemistry for SSTR2 and FGF23 (which is not entirely specific), were any molecular studies performed?

3. Histologic pictures can be improved, particularly including H&E images of the features that are diagnostic of PMT. (I suggest including the Pathology colleague that diagnosed these cases in the study. Without their contribution, this case series would not have happened)

4. In Figure 6, the designation of "Obese spindle cells", although entertaining, is not accurate. Please refer to the WHO Classification of Bone and Soft Tissue tumors for better understanding of the histologic features of this disease.

5. A table summarizing the biochemical studies (serum and urine Calcium, Phosphate, FGF23, etc) in these patients would be interesting and helpful to understand the clinical diagnosis and preoperative biochemical findings.  

6. A table summarizing the demographic features of the patients, tumor location, treatment (excision vs. ablation) and outcomes of all 12 patients would be better for the readers to understand. 

7. There are multiple grammatical/language and typographic errors that make the manuscript difficult to read. Some of these include (but are not limited to) the following:
- Introduction, line 1: "Carcinogenic osteomalacia" - this is not a term I had heard before; I would remove it. 
- Introduction, line 6: "phosphate uromesenchyma" - this is an unfortunate typo. 

Author Response

Dear Editors and Reviewers,

On behalf of my co-authors, we thank you very much for giving us an opportunity to revise our manuscript, we appreciate editors and reviewers very much for their positive and constructive comments and suggestions on our manuscript entitled “Phosphaturic mesenchymal tumors: rethinking from the perspective of clinical diagnosis and surgical treatment”. We have studied reviewers’ comments carefully. Those comments are all valuable and very helpful for revising and improving our paper, as well as the important guiding significance to our study. We have tried our best to revise our manuscript according to the comments, which we hope meet with approval. Revised portion are marked in red in the paper. The main corrections in the paper and the responds to the reviewers’ comments are as follows.

Comments to the Author

Reviewer: 2

  1. I have major issues with the histologic description of these cases. It appears that a Pathologist with experience in bone and soft tissue tumors was not involved in this study (and if a Pathologist made these diagnoses, which they did, they should be included in the study). PMT, although challenging to diagnose, is a well-defined pathologic entity, driven by gene fusions involving FGF1 or FGFR1 in up to 50% of cases. It is typically composed of cytologically bland fibroblasts with a prominent vascular network, and amorphous, "grungy" calcifications. The authors' statement saying "This cohort's pathological findings lacked specificity, showing different histocytological morphology, such as giant cell-change-rich fibrous tissue, cell-rich angiofibroma, non-ossifying fibroma, and osteoblastoma" is problematic. It is true that PMT may have some histologic overlap with these entities, but that does not make its histologic features non-specific. Similarly, the statement "The diagnosed tumor types include giant bone cell tumor, non-ossifying fibroma, fibrosarcoma, osteosarcoma, osteoblastoma, sclerosing hemangioma, angiofibroma, angiolipoma, or other mesenchymal tumors" is simply not true. The authors need to more carefully read references 13-15.

Answer: Thank you very much for taking time out of your busy schedule to review this manuscript. I apologize for many incorrect terms in the manuscript due to our lack of understanding of the pathology of PMT, and thank you very much for your valuable advice to us. Colleagues in the Department of Pathology did make a great contribution to this study. Considering that adding authors is a complicated process, we decided to acknowledge them in the acknowledgements section of the manuscript by obtaining the consent of colleagues in the department of Pathology and all the co-authors of this manuscript. Just as you said, PMT are most fundamentally composed of a hypocellular proliferation of bland neoplastic spindled cells growing in a highly vascular, hyalinized, partially calcified basophilic matrix. However, a single PMT has significant differences in cell, vascular and matrix composition. Therefore, the morphological spectrum of PMT is quite extensive, which undoubtedly explains the reason why PMT is confused with giant bone cell tumor, non-ossifying fibroma, fibrosarcoma, osteosarcoma, osteoblastoma, sclerosing hemangioma, angiofibroma, angiolipoma, or other mesenchymal tumors. According to your suggestion, we have revised the description related to PMT pathology in the manuscript, and we hope that the revised manuscript can get your approval. Thank you again for your valuable suggestions on this manuscript.

  1. Besides immunohistochemistry for SSTR2 and FGF23 (which is not entirely specific), were any molecular studies performed?

Answer: Thank you for your professional suggestion. In addition to immunohistochemistry of SSTR2 and FGF23, we also detected SATB2, ERG, CD56, S-100, etc., but these protein molecules were not expressed in all cases. Unfortunately, in terms of DNA molecular pathology, we did not conduct fluorescence in situ hybridization for FGF1 or FGFR1 gene fusion. However, based on your suggestion, we will definitely add FGF1 or FGFR1 fusion gene detection in future case diagnosis, so as to make more accurate pathological diagnosis of PMT. Thank you again for your valuable suggestions on this manuscript.

  1. Histologic pictures can be improved, particularly including H&E images of the features that are diagnostic of PMT. (I suggest including the Pathology colleague that diagnosed these cases in the study. Without their contribution, this case series would not have happened)

Answer: Thank you for your professional suggestion. We have modified the histological pictures and selected more typical areas for display. Your profound knowledge of pathology has benefited us a lot, and your suggestions have also given us a deeper understanding of the preciseness of academic papers. Thank you again for your valuable suggestions.

  1. In Figure 6, the designation of "Obese spindle cells", although entertaining, is not accurate. Please refer to the WHO Classification of Bone and Soft Tissue tumors for better understanding of the histologic features of this disease.

Answer: Thank you for your professional suggestion. In the picture, we have modified "Obese spindle cells" to "bland neoplastic spindled cells". Thank you again for your valuable suggestions.

  1. A table summarizing the biochemical studies (serum and urine Calcium, Phosphate, FGF23, etc) in these patients would be interesting and helpful to understand the clinical diagnosis and preoperative biochemical findings.

Answer: Thank you for your professional suggestion. We have detected and recorded the biochemical indicators and some hormone levels related to bone metabolism of all patients in this group of cases. However, due to the limitations of the manuscript layout, we only displayed the statistical results, and we uploaded these original data in the form of an supplemental table (Biochemical examination) for peer review. Thank you again for your valuable suggestions.

  1. A table summarizing the demographic features of the patients, tumor location, treatment (excision vs. ablation) and outcomes of all 12 patients would be better for the readers to understand.

Answer: Thank you for your professional suggestion. We recorded and statistically analyzed the demographic characteristics, course of disease, lesion location, lesion size, treatment methods and follow-up time of all patients in this group. However, considering the limitation of manuscript layout, we only showed the statistical results of these factors, and we uploaded the original data in the form of an supplemental table (General information) for peer review. Thank you again for your valuable suggestions.

  1. There are multiple grammatical/language and typographic errors that make the manuscript difficult to read. Some of these include (but are not limited to) the following:

- Introduction, line 1: "Carcinogenic osteomalacia" - this is not a term I had heard before; I would remove it.

- Introduction, line 6: "phosphate uromesenchyma" - this is an unfortunate typo.

Answer: Thank you for your professional suggestion. According to your suggestions, we have revised the wording errors and grammar errors of the manuscript, and made professional language modifications to better meet the publication requirements of the magazine. Thank you again for your valuable suggestions.

Once again, thank you very much for editor’s and reviewers’ positive and constructive comments. Your suggestions make us progress continuously. We hope that our studies will provide further experience and evidence for clinical diagnosis and surgical treatment of PMT.
